# Molecular Mechanisms of Acute Organophosphate Nephrotoxicity

**DOI:** 10.3390/ijms23168855

**Published:** 2022-08-09

**Authors:** Vladislav E. Sobolev, Margarita O. Sokolova, Richard O. Jenkins, Nikolay V. Goncharov

**Affiliations:** 1Sechenov Institute of Evolutionary Physiology and Biochemistry, Russian Academy of Sciences, Thorez 44, 194223 St. Petersburg, Russia; 2Leicester School of Allied Health Sciences, De Montfort University, The Gateway, Leicester LE1 9BH, UK or

**Keywords:** nephrotoxicity, organophosphates, acute poisoning, animal models, albumin, endothelium, morphology, kidney, glycosaminoglicans

## Abstract

Organophosphates (OPs) are toxic chemicals produced by an esterification process and some other routes. They are the main components of herbicides, pesticides, and insecticides and are also widely used in the production of plastics and solvents. Acute or chronic exposure to OPs can manifest in various levels of toxicity to humans, animals, plants, and insects. OPs containing insecticides were widely used in many countries during the 20th century, and some of them continue to be used today. In particular, 36 OPs have been registered in the USA, and all of them have the potential to cause acute and sub-acute toxicity. Renal damage and impairment of kidney function after exposure to OPs, accompanied by the development of clinical manifestations of poisoning back in the early 1990s of the last century, was considered a rare manifestation of their toxicity. However, since the beginning of the 21st century, nephrotoxicity of OPs as a manifestation of delayed toxicity is the subject of greater attention of researchers. In this article, we present a modern view on the molecular pathophysiological mechanisms of acute nephrotoxicity of organophosphate compounds.

## 1. Introduction: Organophosphates (OPs), Their Sources, Routes of Ingestion, and Principal Targets

Organophosphates (OPs) are toxic chemicals produced by an esterification process and some other routes. They are the main components of herbicides, pesticides, and insecticides and are also widely used in the production of plastics and solvents [1]. Acute or chronic exposure to OPs can manifest in various levels of toxicity to humans, animals, plants, and insects. OPs containing insecticides were widely used in many countries during the 20th century, and some of them continue to be used today. In particular, 36 OPs have been registered in the USA, and all of them have the potential to cause acute and sub-acute toxicity [2]. Organophosphorus pesticides are divided into diethyl (parathion, paraoxon, quinalphos), dimethyl (monocrotophos), and S-alkyl (profenophos, protiophos) classes. Most organophosphorus pesticides are thioates, with a sulphur atom double bonded to a phosphate (parathion, quinalphos, protiophos), which must be converted to the active oxone (parathion to paraoxone). Some organophosphate pesticides, oxones (profenophos, protiophos), do not need to be activated; they are able to inhibit acetylcholinesterase immediately after absorption [3]. Organophosphate pesticides are highly toxic compounds that are still widely used in agriculture worldwide. In the 1970s, the EPA (Environmental Protection Agency) estimated that there were 3000 hospitalisations annually in the United States due to insecticide poisoning [4], and mortality was as high as 50% in children and 10% in adults. In 1983, according to the American Association of Poisons Control Centers in the USA, 77,000 cases of insecticide exposure were registered, 33,000 of which were related to organophosphorus compounds [5]. Pesticide poisonings are most common among agricultural and industrial workers, and among children [6]. Suicidal self-poisoning by these readily available agents has been a serious problem in developing countries, such as Sri Lanka, where they accounted for up to 90% of reported cases [7,8]. In the United States, 13,348 poisonings with OPs were registered in 1999, a quarter of which were treated in medical institutions, with five fatalities [9]. In the 21st century, pesticide OP poisoning remains an important clinical problem in rural areas of the developing world, killing up to 200,000 people each year according to some estimates [3]. In the second decade of the 21st century, suicidal self-poisoning of OPs, including by parenteral route, also continues to feature in medical reports in many countries [10,11,12]. Currently available antidotes for the treatment of severe OPs poisoning are symptomatic; do not reduce intoxication in the body; and have limited ability to prevent long-term damage to brain, liver, and kidney tissues [13].

Human and animal exposure to OPs occurs in several ways: through contaminated water, air, food, and contaminated environment, including through the skin [14,15]. Penetrating into the body, OPs damage various tissues, including the brain and liver, which is associated with acute and chronic diseases [16]. The population is exposed to pesticide OPs mainly through the digestive tract [17,18,19,20]. Organophosphate compounds are also effectively absorbed by inhalation. The dermal penetration of OPs and their subsequent systemic absorption vary depending on the specific agent. 

There is considerable variation in the relative absorption of OPs by different routes [2]. For example, the LD50 of parathion for oral administration in rats is 3–8 mg/kg, which is quite toxic and essentially equivalent to dermal absorption at an LD50 dose of 8 mg/kg [21,22]. On the other hand, the toxicity of some OPs (e.g., fosalone) at dermal route of administration is much lower than at oral route, with LD50 for rats being 1500 mg/kg and 120 mg/kg, respectively [22]. According to other data, the LD50 for parathion in rats when administered per os (p.o.) is 6–7 mg/kg and when administered intramuscularly (i.m.) is only 500–600 µg/kg. The human toxicity of parathion, in contrast to rodents, in both oral and parenteral routes of ingestion is comparable being 50–200 mg/70 kg and 2.8–3 mg/kg respectively [23]. 

To some extent, the occurrence of poisoning depends on the rate of absorption of the pesticide. The breakdown of OPs occurs mainly by hydrolysis in the liver, and the rate of hydrolysis varies widely from one compound to another. In those OPs whose breakdown is relatively slow, there may be significant temporary accumulation in fat [2]. Some OPs, such as diazinon, fentione, and methyl parathion, have significant lipid solubility, allowing them to accumulate in fat with delayed toxicity due to late release [24,25]. Delayed toxicity can also occur atypically with other OPs, particularly dichloropentione and demton methyl [26]. 

Damage to the kidneys and their function after exposure to OPs compounds, accompanied by the development of clinical manifestations of poisoning, was considered a rare phenomenon in the early 1990s [27]. At that time, only three publications mentioning nephrotoxic effects after OPs poisoning in humans had been published [28,29,30]. In the 21st century, the nephrotoxicity of organophosphorus pesticides has been the subject of greater attention. In in vivo experiments on laboratory animals, nephrotoxicity of OPs has been detected for fenthion [31,32], diazinon [33], malathion [34,35], chlorpyrifos [36], dichlorophos [37], metamidophos [38], quinalphos [39], and methyl parathion [40]. At the same time, there has been an increasing number of publications on clinical cases of human poisoning with OPs accompanied by nephrotoxic effects, including acute kidney injury (AKI) and acute renal failure (ARF) [41,42,43]. ARF is one of the problems that manifests itself in the clinical follow-up of patients and is responsible for the increased mortality in AKI poisoning [41,43]. Individuals exposed to OPs were found to have 6.17-times higher risk of developing ARF, after adjusting for age, gender, and comorbidities [42]. Various mechanisms have been proposed to explain the development of ARF after OP intoxication; however, due to insufficient experimental data, knowledge on this issue is limited. On the basis of the experimental data presented above, the proximal and distal tubules as well as certain components of the renal calf are currently considered to be the most vulnerable elements of the nephron (Figure 1). 

AKI is a likely consequence of OPs’ poisoning. A retrospective cohort study published in 2015 of 8924 patients aged over 20 years showed that the overall incidence of AKI was higher than in controls (4.85 versus 3.47/1000 person-years). After adjustment for age, sex, comorbidity, and interaction conditions, patients with OPs poisoning had a 6.17-fold higher risk of AKI compared to the comparison group. Patients with very severe OP poisoning were associated with a significantly increased risk of developing AKI [42].

Organophosphate pesticides may also contribute to the progression of chronic kidney disease (CKD) [45]. Results from a recent study of 618 children with CKD showed that urinary metabolites of dialkylphosphate (DAP) are associated with subclinical renal damage, which may signal the possibility of future clinical manifestations [46]. All these facts indicate the need for a thorough study of the pathophysiological mechanisms of nephrotoxicity of OPs in humans and animals. One of the main tools of such studies is the use of toxicological models with laboratory animals.

### Animal Models of Acute Organophosphate Toxicity

Acute toxicity is generally defined as an adverse change occurring immediately or within a short time after a single, repeated, or short exposure to a substance, or as adverse effects occurring within a short time after administration of a single dose of a substance or several doses administered within 24 h [47,48].

The effects of OPs have been studied in different animal species, mainly rodents, primates, aquatic organisms, and pigs, using different routes of administration: intraperitoneal, intravenous, intramuscular, subcutaneous, contact, inhalation, and oral. Several excellent reviews present this information in more detail [49,50,51]. 

The translational ability of data from preclinical toxicological studies depends on several factors, including the adequacy of the animal model [52]. The clinical manifestations of acute human exposure to OPs can be accurately reproduced in rodents and non-human primates. These manifestations include acute cholinergic crisis in addition to signs of neurotoxicity that develop long after OPs exposure, in particular chronic neurological deficits consisting of anxiety-related behaviour and cognitive deficits, structural brain damage, and increased frequency of slow electroencephalograms. Guinea pigs and non-human primates, such as humans, have low levels of circulating carboxylesterases that metabolise and inactivate OPs, and therefore these species are recognised as the most suitable animal models to study OPs intoxication [52].

Rats, as one of the most frequently used animal model species, continue to be used in toxicological experiments with OPs [49,52,53]. This animal species has specific physiological features, in particular the presence of high carboxylesterase (CE) activity in blood plasma. Mammalian CEs belong to a multigenic superfamily with broad substrate specificity, catalysing the hydrolysis of esters, thioesters, amide-containing xenobiotics, and endogenous compounds, including fatty acid esters [54,55]. In contrast to rodents and hares, plasma of humans, monkeys, and hornbills does not contain CE [56]. To this end, the inhibition of rodent blood plasma CE activity may improve the adequacy of experimental models for the study of OPs mechanisms of action. Earlier, we proposed two models of acute poisoning of rats by OPs, with preliminary inhibition of CEs by administration of 2-(o-cresyl)-4H-1,3,2-benzodioxaphosphorin-2-oxide (CBDP) or fractional administration of a total dose of OPs under study [57]. The models allow effective assessment of various aspects of OP toxicity manifestation in rats, comparable to human poisoning [58,59,60,61]. It is advisable that choice of animal species and poisoning model should take into account the potential effect of different OPs on the target organs. In our studies, it was shown that some OPs, such as paraoxon, show signs of nephrotoxicity even at a single exposure, regardless of prior inhibition of CE activity [61]. 

## 2. Molecular Pathophysiological Mechanisms of Organophosphate Nephrotoxicity

The pathophysiological mechanisms of the development of the principal clinical syndromes in OP poisoning include three main reactions: cholinergic with anticholinesterase effects, membrane toxic, and pro-oxidant. The cholinergic effects of OPs consist of anticholinesterase and cholinoreceptor effects, but also include non-cholinergic mechanisms of action of OPs: glutamate, adrenergic, etc. These reactions provide microcirculatory disorders, hypoxia, cytotoxic effects, and cell death, as well as immunopathological disorders. Molecular signalling pathways leading to the survival or death of renal cells after exposure to OPs are poorly understood. Some molecular mechanisms responsible for the implementation of nephrotoxic effects of OPs are discussed in more detail below.

### 2.1. Serum Albumin

In terms of nephrotoxicity of OPs, three molecular mechanisms are of most interest in the interaction of these compounds with serum albumin (SA): (1) binding of OPs by albumin; (2) transport/interaction of OP-SA complexes and metabolites with kidney tissues and cells; (3) protective effects of albumin against OPs. Each of these mechanisms are considered in more detail below. 

Albumin is the main protein in mammalian blood, where its concentration is in the range of 500–700 µM. It is synthesised in the liver at a rate of about 0.7 mg per hour (i.e., 10–15 mg per day); the half-life of bovine serum albumin (BSA) is 19–20 days [62]. The structure of albumin is conservative in all mammals: the molecule consists of three homologous domains, each of which consists of 10 helices and can be divided into two subdomains (A and B) containing six and four helices, respectively, with both subdomains connected by a long loop [62]. The human serum albumin (HSA) molecule is formed by a single polypeptide chain consisting of 585 amino acid residues. In albumin of other species, the polypeptide chain length can vary; in particular, BSA contains 584 amino acid residues, and rat serum albumin (RSA) consists of 583 residues [63]. HSA has a molecular weight of 66,348 Da and consists of three homologous domains numbered I, II, and III [64]. Each domain includes subdomains A and B, which share common structural motifs. The two main regions responsible for ligand binding to SA are known as Sudlow’s site I and II, located in subdomains IIA and IIIA, respectively [65]. Albumin is encoded by a single gene that is expressed in a co-dominant manner with transcription and translation of both alleles. The human albumin gene is located on the long arm of chromosome 4 at position q13 [66].

#### 2.1.1. Binding of OPs to Albumin

The binding of OPs of pesticides to plasma proteins is one of the many factors affecting their distribution and excretion. There are several different transport proteins in plasma, but only albumin is able to bind a wide range of xenobiotics reversibly and with high affinity [67]. There are also cooperative and allosteric modulating effects on the interaction of albumin with various substances, most prominent in multimeric macromolecules [68,69]. In this regard, ligand binding at one site may affect the binding efficiency at another site. Similar conformational changes occur in the albumin molecule following the binding of a number of endogenous compounds such as bilirubin [70], urea [71], estradiol [72], and glucose [73]. Exogenous compounds can also exert allosteric effects. For example, lorazepam binding at the Sadlow II site alters the binding efficiency of warfarin at the Sadlow I site [74], and tenoxicam binding at the Sadlow I site enhances diazepam binding at the Sadlow II site and vice versa [75]. 

The structure of OPs plays an important role in determining its affinity for albumin in the hydrolysis of OPs metabolites [76]. Some OPs (ethyl parathion) have high affinity to human and bovine serum albumin. Results show that the main binding site of ethyl parathion on albumin is near tryptophan residue 214 of human SA and 212 of bovine SA [67].

The (pseudo)esterase activity of albumin against Ops—phosphoric or phosphonic acid esters—is also of great importance in toxicology. The sites of formation of covalent adducts of OPs with albumin were established recently [62]. In addition to (pseudo)esterase activity, albumin exhibits peroxidase activity against lipid hydroperoxides and many others. Despite the obvious success of modern analytical methods and the desire of some scientists to use them to prove the absence of true esterase and other enzymatic activities in albumin, the results obtained do not solve the problem unambiguously. So far, the mechanisms of interaction of various esters and other compounds with albumin remain unsolved [62]. 

In in silico experiments investigating the interaction of albumin with OP, the influence of HSA redox status on their interaction with paraoxon was studied [77]. According to the data obtained, the redox status of Cys34 has no significant effect on the possibility of esterase reactions at Sudlow site I. The activity of Sudlow I site of HSA is independent of the redox status of cysteine. However, modification of cysteine changed the conformation of Sudlow I site of HSA and the position of the paraoxon molecule within the site. Oxidation of albumin had almost no effect on the Sudlow II site conformation of HSA, nor on the position of the ligand in this site, nor on the affinity of the site for the paraoxon [77].

Saturation transfer difference NMR experiments combined with molecular docking studies revealed the binding interaction of chlorpyrifos, diazinon, and parathion in solution, confirming the binding of OPs-BSA complexes, while isothermal titration calorimetry experiments showed high-affinity binding of these complexes via a non-covalent interaction [78]. Several irreversible and productive OP binding sites for HSA and BSA were also demonstrated [79].

Considering the above facts in terms of nephrotoxicity of OPs, it should be noted that there are currently no published studies confirming the accumulation of OP-bound complexes with albumin in the kidneys. For some OPs, such as quinalphos, there is evidence of the presence of protein traces in glomerular basement membrane (GBM) after exposure, but their belonging to OP-SA complexes has not been proved [39].

#### 2.1.2. OP-SA Complexes in the Kidneys: More Questions Than Answers

Hypothetically, OP-SA complexes within circulating blood are distributed throughout the body and may interact with tissues that have receptors that bind albumin. One such receptor was identified in the early 1990s as secreted-albumin-binding protein (SPARC) [80]. This protein, also known as osteonectin (BM-40), is secreted by several cell types: endothelial cells and vascular smooth muscle cells, as well as in skeletal muscle; fibroblasts; and testicular, ovarian, pancreatic, and some tumour cells [81]. However, almost nothing is known about the participation of this protein in the processes of interaction of OPs with albumin and their complexes. 

The albumin-binding proteins cubilin and megalin identified in kidneys are of much greater interest [82,83]. The substrate for these proteins is both native and probably modified albumin. Kubulin is found in proximal tubule cells as well as intestinal, placental, and visceral cells of the yolk sac. Megalin is also present in proximal tubule cells, intestinal absorbing cells, placenta, visceral cells of the yolk sac, thyrocytes, ciliary epithelial cells, lung cells, parathyroid cells, endometrium, oviduct, inner ear cells, and epithelial cells of the testes appendage [66]. Megalin, a 600 kDa transmembrane glycoprotein, can bind various ligands [84], including albumin, vitamin-binding proteins, carrier proteins, lipoproteins, hormones, drugs, enzymes, and immune-related proteins. Megalin can also act as a membrane anchor for cubilin, a 460 kDa glycoprotein on the cell surface that does not have an explicit transmembrane domain and glycosylphosphatidylinositol (GPI) anchor [85]. 

Reabsorption of filtered proteins in the kidney occurs by receptor-mediated endocytosis in the proximal tubules. The receptors responsible for mediated reabsorption have been found to be albumin-binding cubilin and megalin [86,87]. The megalin/cubilin complex is responsible for receptor endocytosis and rescue of albumin from renal excretion. In proximal tubule cells, this complex functions as a scavenger receptor and is the main receptor for renal albumin reabsorption [83,88,89]. Megalin and cubulin are among the most common proteins in the urine of patients with microalbuminuria in type 1 diabetes [90].

Another factor regulating albumin reabsorption in the kidney is the neonatal Fc receptor (FcRn), a heterodimer composed of the major histocompatibility complex (MHC) class I heavy chain and α2-microglobulin light chain. FcRn is widely expressed in adult cells, including endothelial cells and kidneys [91,92]. In the kidneys, FcRn is expressed on the surface of podocytes and in the brush border of proximal tubular cells. FcRn binds albumin (as well as IgG) depending on pH, with greater affinity at acidic pH (<6.5) and less at pH ≈ 7.4 [93,94]. This property is important as it makes FcRn capable of rescuing IgG and albumin from acidic endosomes, preventing their lysosomal degradation and increasing their lifespan [95]. Recently, it has been suggested that FcRn is involved in an important mechanism of transcytotic albumin extraction in proximal tubule cells [96,97], although luminal pH does not promote albumin binding to FcRn [98]. It is therefore assumed that the megalin/cubilin complex binds albumin at the cell surface and FcRn binds albumin in endosomes at low pH. The vesicles containing FcRn-bound albumin fuse with the basolateral membrane, releasing the ligand, and the free albumin is digested lysosomally. Figure 2 shows a schematic representation of this hypothesis [85].

On the basis of the above facts, changes/disruptions in OP-related albumin reabsorption processes, including those on the regulatory side of the megalin/cubilin and FcRn complexes, can be assumed. Unfortunately, it is currently unknown whether studies in this direction have been conducted. In this regard, many more questions remain than answers regarding the molecular pathophysiological mechanisms of the effects of OPs in OP–SA complexes on renal cells. Research in this direction would be very promising.

#### 2.1.3. Protective Effects of Albumin against OPs

Several studies have suggested a protective function of plasma albumin in OP poisoning. For example, in blood samples obtained from patients who died as a result of OP poisoning with malathion, its observed concentration was lower than expected [99]. The researchers attributed the decrease in OP concentration to two mechanisms. The first was the ability of malathion to be degraded by the esterase activity of HSA. As the most abundant plasma protein, HSA serves as a carrier protein for many endogenous and exogenous compounds in the bloodstream. It has been suggested that HSA acts as a carboxylesterase in postmortem blood. Another mechanism by which HSA is involved in postmortem malathion degradation has also been found, namely, the direct interaction of malathion with amino acid residues in HSA [99]. 

In a retrospective study of 217 patients poisoned with OPs, on admission hypoalbuminemia (albumin < 3.5 g/dL) was detected in 18.4% of patients [100]. The group of patients with hypoalbuminemia had a more complicated clinical course and a higher mortality rate than the groups with normoalbuminemia and hyperalbuminemia. SA concentration correlated with CRP level on admission, but not with body mass index in patients with OP poisoning. Moreover, changes in SA concentration during the first 24 h also correlated with changes in BuChE activity in patients with phenythrothion poisoning. SA concentration on admission was independently associated with mortality and was independent of inflammation and nutritional status in OP poisoning. This result may imply that albumin is involved in a protective mechanism against OPs toxicity and that this involvement may contribute to serum albumin-associated mortality risk. However, these findings need further confirmation [100]. 

Given the ability of albumin to act as a biological trap capable of absorbing free OPs, the use of fresh frozen plasma or albumin has been suggested as a treatment for acute cases of poisoning. However, despite a significant increase in pseudocholinesterase levels with their use, this study did not show favourable trends in clinical outcomes when fresh frozen plasma or albumin was used [101].

### 2.2. Oxidative Stress

In the pathophysiology of multiple effects of OP poisoning, hypoxia makes a great contribution. The main causes are airway obstruction and bronchial spasm; changes of respiratory muscle activity—fasciculations, hypertonicity, and paralysis [102,103,104]; microcirculatory disorders [105,106], including erythrocyte aggregation and stasis (Figure 3); heart muscle dysfunction—arrhythmia, hyper- or hypodynamic response [102,107]; depressed respiration; and suppressed oxygen utilisation by tissues [103,104,105]. Hypoxia itself can activate the process of lipid peroxidation (LPO) and disrupt biochemical processes of tissue respiration.

Hypoxia and multi-organ disorders, especially neurotoxic reactions, significantly contribute to the development of oxidative stress during OP intoxication, coupled with inhibition of the antioxidant system [108]. Reactive oxygen species (ROS) contribute to respiratory and renal failure [109], as well as to development of delayed effects of acute OP intoxication. Although different OPs have different spectra of toxic effects, including the prooxidant indices [110], nevertheless, there are also common and non-specific prooxidant effects. 

It is now accepted that one of the main pathophysiological mechanisms of OP toxicity is a change in the oxidant–antioxidant balance in tissues and oxidative stress, with depletion of glutathione and increased lipid peroxidation—the process of oxidation of lipid layer of cell membranes [109,110,111,112]. LPO is a destructive, self-sustaining chain reaction that produces malondialdehyde (MDA) as the end product [113], which can be measured by the thiobarbiturate reactive substances test (TBARS) [114,115]. The results of many published studies confirm the association of OP poisoning with oxidative stress in rats [116,117] and humans [118,119]. LPO under the influence of OPs was detected in rat brain [120] and human erythrocytes [116]. OP-induced seizures associated with oxidative stress have been reported [121]. For example, malathion induces a state of oxidative stress in renal tissues, mainly due to an increase in LPO and depletion of reduced glutathione (GSH) glutathione peroxidase (GPx), catalase (CAT), and superoxide dismutase (SOD) [122,123,124,125].

Acute tubular necrosis developing after OP poisoning is also associated with ROS and LPO [126]. The level of oxidative stress and the degree of renal dysfunction in OP poisoning may have a dose-dependent character, which was shown in a rat model of 8-week diazinon poisoning. Thus, the most characteristic picture of induced oxidative stress/renal failure, accompanied by an increase in blood creatinine and urea nitrogen, was observed at high doses—15 and 30 mg/kg b.w. [109]. Paraoxon (POX) also induces ROS production and oxidative stress in a dose-dependent manner. The induction of oxidative stress in POX-treated rats occurs in the following order: brain > liver > heart > kidney > spleen [127].

Most researchers acknowledge the fact that oxidative stress is a concomitant factor in the lethal outcome of the OP-induced poisoning, in addition to AChE inhibition [128]. In sub-chronic or chronic exposure to OPs, oxidative stress is considered by many authors to be the main mechanism of toxicity of these compounds in the context of in vitro and in vivo animal studies, as well as in clinical studies [129]. Anyway, the role of oxidative stress in the pathophysiology of OP intoxication is actively discussed, but whether it plays a direct toxic role is currently unclear [130]. It should also be noted that each OP has a unique toxicity profile, and the hypothesis/concept cannot be generalised to all compounds. For example, ingestion of dichlorfos does not cause marked oxidative stress and probably does not play a major role in the pathological processes [128].

Human or animal contact with OPs causes neurotoxic changes expressed in overstimulation of cholinergic and glutamatergic systems. This is accompanied by increased formation of ROS and oxidative tissue damage, with initiation of molecular mechanisms related to LPO, where ROS can enhance intracellular oxidative processes. Several haematological, neurological, metabolic, and genotoxic diseases may be linked to the effects of LPO induced by pesticide intoxication [131,132]. The molecular signalling pathways leading to cell survival/death after exposure to OPs are not yet fully understood, although protein oxidation, together with activation of signalling pathways and cell apoptosis, are described [133].

Exposure to OPs causes hyperglycemia, which leads to increased non-enzymatic glycation by binding glucose or by-products to protein amino groups and forming complex compounds, advanced glycation end products (AGE), which alter protein structure and function. Glycated proteins activate specific receptors (RAGE) and cause intracellular oxidative stress [129]. Thus, hyperglycaemia is one of the mechanisms of oxidative stress in OP intoxication [134]. On the other hand, there is evidence that antioxidants need glucose, as mentioned above, for their action against ROS [129,135].

Oxidative stress induced by exposure to OPs can stimulate the mitogen-activated protein kinase (MAPK) signalling pathway in tissues, which play an important role in the regulation of gene expression in all cellular activities, after receiving extracellular signals in the form of mitogenic or genotoxic (mutagenic) effects, as well as in response to cytokines. There are three subfamilies of these serine/threonine kinases in mammals, including extracellular kinases (ERKs), c-Jun N-terminal kinases (JNKs), and p38-MAPKs, which can be activated by different types of OPs. Activation of ERK, JNK, and p38 kinases in response to oxidative stress through ROS production has both survival-promoting and pro-apoptotic effects. The oxidative effect of organophosphorus pesticides is mediated specifically by MAPK signalling and is caused by an imbalance between the oxidative and antioxidative systems, leading to a disruption of antioxidant protection [133].

The nephrotoxicity of OPs may be mediated by the MAPK/ERK signalling pathways (Figure 4). Recent studies have shown that these pathways play an important role in nephrogenesis and lead to the differentiation of nephrogenic mesenchyme. Moreover, an imbalance in MAPK signalling pathways causes a number of diseases, including cancer. Renal cell development is driven by MAPK signalling and an imbalance in MAPK signalling pathways, especially in nephron precursors, and leads to impaired nephron differentiation. Tris-(2-chloroethyl)-phosphate (TCEP) at concentrations of 0.01 and 1 mg L^−1^ was shown to significantly increase JNK phosphorylation and decrease Bcl-2 and ClAP-2 expression. TCEP also increased the expression of caspase-3 and caspase-9 in primary cultures of proximal renal tubule cells [133].

Two main systems are known to counteract ROS-induced damage in humans and animals. One is an enzymatic system consisting of SOD, CAT, GPx, and glutathione reductase (GR) [136,137,138]. Another, non-enzymatic antioxidant system consists of GSH, vitamins C and E, β-carotene, uric acid, ceruloplasmin, and ubiquinone [139,140]. Each of these antioxidant systems has a certain activity, and they function synergistically [129]. In antioxidant action, GPx uses GSH, and in this reaction, GSH is converted to the oxidised form of glutathione (GSSG). Cells use the enzyme GR to recycle GSSG into GSH [117,141,142]. GR needs NADPH as a substrate for the reduction of oxidised form of glutathione to its reduced form. The source of NADPH is the pentose phosphate pathway with glucose-6-phosphate dehydrogenase (G6PD) as a rate-limiting enzyme whose important function is the reduction of NADP+ to NADPH. G6PD needs glucose as a substrate for its activation [129,134]. Oxidative stress reduces the stock of GSH [137,141,142]. In addition, the level of GSH may be reduced due to its participation in conjugation reactions or reduced ability of cells to regenerate GSH. The effects of oxidative stress are manifested by increased concentration of MDA, which is a marker of LPO, as well as increased levels of ROS [143,144].

SOD and MDA levels are generally considered as markers of antioxidant status and oxidative stress [113]. Elevated MDA levels in patients with OP poisoning who fail to survive probably reflect accelerated LPO, cell damage, and death [113]. Given the importance of oxidative stress in the pathogenesis of OP poisoning, the efficacy of natural or synthetic antioxidants in therapy has been investigated. It is emphasised that animal experiments, and rat models of poisoning in particular, do not fully replicate the clinical situation in humans. For this reason, clinical trials are needed to study the effectiveness of antioxidants in protecting against OPs toxicity [129].

A review study summarising the effectiveness of antioxidants in acute OPs poisoning reported that traditional markers of lipid peroxidation may not change much. In addition, a high variability in the results and efficacy of different antioxidants has been observed [130]. Oxidative damage to lipids and proteins was reported in all 11 human studies. Eight out of nine studies reported a mild increase in MDA. In two case–control studies, the MDA concentration in erythrocyte membranes was 380% and 160% higher than in controls, and the MDA concentration in plasma was ≈63% higher than in controls in three case–control studies [130]. In another study after acute parenteral poisoning of rats with POX at doses of 0.7 mg/kg and 1 mg/kg, renal SOD and catalase activities were significantly increased compared to controls, and GSH levels were significantly decreased. No significant changes in GST, LDH, and MDA activity were observed [145]. 

Given the increased levels of oxidative stress due to OP poisoning, antioxidants have been suggested as an adjunct to standard therapy. However, no significant evidence of benefit of their use for survival after acute OP intoxication was found [128]. Nevertheless, various stabilisers that reduce oxidative stress levels are likely to have some beneficial effects. For example, L-carnitine administration improved antioxidant status and reduced the total dose of atropine required to treat patients with acute OP poisoning, provided that at the time of patient admission MDA and GSH levels, total serum antioxidant capacity, and pseudocholinesterase enzyme activity had no significant differences between the two study groups [146]. 

A study evaluating the efficacy of three non-enzymatic antioxidants—*N*-acetylcysteine, glutathione, and ascorbic acid—in acute intoxication of adult male Wistar rats with POX concluded that the tested non-enzymatic antioxidants were not useful in acute toxicity to improve survival [128]. It should also be born in mind that different OPs have individual toxicity dynamics and therefore other compounds should be tested in this regard. It has therefore been suggested that oxidants produced during acute intoxication with POX are not absorbed by these antioxidants, as they may be under the control of enzymatic antioxidants. Nevertheless, on the basis of the literature, it can be suggested that the use of antioxidants may be useful in chronic OPs exposure, which causes various pathophysiological conditions due to oxidative stress [128].

The effect of N-acetylcysteine on melastatin transient receptor potential 2 (TRPM2) channel expression in rat kidney and liver tissues after experimental malathion intoxication was investigated [147]. TRPM2 is a calcium-permeable channel that is sensitive to oxidative stress and is thought to contribute to calcium influx associated with neurodegenerative diseases. In renal tissue, MDA levels, apoptosis, and TRPM2 immunoreactivity were significantly increased in the malathion and malathion + *N*-acetylcysteine groups compared to controls. OPs intoxication had no effect on MDA levels, apoptosis, and TRPM2 immunoreactivity in rat liver during the acute period, while in kidney tissue MDA, apoptosis and TRPM2 immunoreactivity were significantly increased after malathion administration [147]. In another study, TRPM2 deficiency was shown to reduce chronic unpredictable stress induced by ROS and calpain activation and prevent aberrant hyperactivation of cyclin-dependent kinase 5 (Cdk5) [148].

### 2.3. Endothelial Damage

The endothelium is an internal cell monolayer lining blood and lymphatic vessels. The endothelium is not only a channel for blood transport and a barrier regulating vascular permeability and immune cell diapedesis at sites of inflammation, but also a regulator of many important physiological functions (Table 1). The kidney, as a highly vascularised organ, is characterised by a great diversity of endothelial cells (ECs), including medium and large vessel ECs, glomerular ECs (GECs), and peritubular capillary cells [149].

GECs are involved in glomerular filtration, while peritubular capillary ECs are involved in tubular secretion and reabsorption processes. Both GECs and peritubular ECs are fenestrated [150], and these cells are crucial for the permissiveness of the glomerular filtration barrier and for the efficient passage of large volumes of fluid and urine formation. GECs are covered by a thick glycocalyx enriched with negatively charged proteoglycans, predominantly heparan and chondroitin sulphates, which form a network with glycosaminoglycans [149]. Glycocalyx contributes to the regulation of vascular permeability and fluid balance and repels blood cells from the vascular wall [151]. Densely packed hyaluronic acid (HA) attaches glycocalyx to GBM and fills endothelial fenestrations, preventing the passage of albumin [152]. The thickness of the glycocalyx normally exceeds the size of cell adhesion molecules such as intercellular adhesion molecule 1 (ICAM1), vascular cell adhesion protein 1 (VCAM1), P-selectins, and E-selectins that are expressed by EC, which prevents cell adhesion to the endothelium [149]. 

Peritubular capillary ECs have a thin stroma, and cell fenestrations are covered by a thin layer formed of glycoproteins organised into radial fibrils, which modulates the screening properties of these ECs [149,153,154]. These ECs lie on the basal membrane, on the opposite side of which are pericytes, which provide capillary constriction to regulate medullary and cortical blood flow; they also regulate capillary permeability, participate in angiogenesis, and are the source of myofibroblasts in renal fibrogenesis [149,155,156]. Peritubular microvascular ECs also provide a niche for resident organ stem cells, which are in close proximity and benefit from the paracrine effects of the angiocrine factors they secrete. This angiocrine signalling is organ-specific and plays a crucial role in tissue repair and prevention of scarring and fibrosis [149,157].

A number of adverse pathophysiological events occur in endothelial cells under the influence of OPs, which have been shown in in vitro experiments in the human umbilical vein EC (HUVEC) model. In particular, POX in a concentration- (36.3 nmol/L~36.3 μmol/L) and time-dependent manner was found to inhibit endothelial monolayer permeability in HUVEC. Exposure to POX simultaneously leads to a decrease in SOD activity and nitric oxide (NO) content and an increase in MDA content in both aortic tissue and cell culture medium [158], and consequently increases permeability of the endothelial cell monolayer. 

It should be noted that compared to HUVECs, which are considered the classical EC model, GECs produce more fibrinolytic factors at rest. Moreover, the fibrinolytic properties of GECs are enhanced when activated by TNF or LPS, whereas HUVECs become more pro-thrombotic [159]. In contrast to differences in the expression of thrombotic factors, weak differences exist between GECs and HUVECs in the expression levels of adhesion molecules, complement factors, complement activation capacity, and permeability at rest and after activation by TNF, IFNγ, or IL-1, or short-term exposure to haem [149,160,161,162,163]. Furthermore, GECs have a different metabolic profile than HUVECs at rest and after exposure to factors such as Stx and/or TNF [164]. Exposure to Stx and TNF on GECs strongly decreases NAD metabolism, leading to depletion of the energy substrate acetyl CoA and the antioxidant glutathione [149]. These findings suggest that Stx and TNF make GECs particularly susceptible to oxidative stress, which logically leads to the assumption that pro-oxidative products of haemolysis, such as haemoglobin, haem, and erythrocyte microvesicles [165], are particularly dangerous for renal microvessels and enhance cellular damage, complement activation, and thrombosis in the context of renal thrombotic microangiopathy (TMA) [149,166,167].

Unfortunately, there is currently very little information on the effects of OPs on EC kidney function under in vivo conditions. There are sporadic publications in which these effects have been considered in an acute poisoning model in laboratory animals. In particular, after intraperitoneal injection of methamidophos at a dose of 30 mg/kg into rats, no ultrastructural effects of acute poisoning were observed in the kidneys, including no signs of EC damage [38]. At the same time, the authors of the study acknowledge that models using other OPs at different doses may lead to different results. 

Hypothetically, the possible mechanisms of renal EC damage under the influence of acute OP poisoning can in some cases correlate with the processes occurring in diabetic nephropathy. This is due to the hyperglycaemia that develops after acute OP poisoning. Hyperglycaemia and the binding of advanced AGEs to their RAGE receptor and oxidative stress contribute to glycocalytic destruction by increased heparinase production, reduced heparan sulphate synthesis, apoptosis of glomerular endothelial cells (GECs), and reduced NO synthesis. Progressive loss of fenestrated GECs and podocyte stem detachment increase glomerular permeability. An excellent scheme of this process is presented in two extensive reviews [149,168].

Depending on the duration of poisoning, the chemical structure, and dose of OPs, the damage to the EC kidney, as well as other endothelial cells, can be considered as a sequence of early and late events developing under the influence of oxidative stress (Figure 5) [169].

### 2.4. Glycosaminoglycans and Organophosphate Nephrotoxicity 

#### 2.4.1. Basement Membrane Glycosaminoglycans

Glycocalyx, secreted by endothelial cells and covering the luminal surface of blood vessels, consists of a tightly linked network of GAGs, including heparan and chondroitin sulphates, keratansulfate, type IV collagen, laminins, fibronectin, hyaluronic acid, and other extracellular matrix proteins. Glycocalyx is associated with endothelial cells through membrane-associated CD44 and the proteoglycans syndecan and glypican [170,171]. The orozomucoid, secreted by endothelial cells, imparts a negative charge to glycocalyx, which inhibits BBB permeability to negatively charged plasma proteins [172]. Apparently, its main barrier function is that it is a molecular sieve, as the diffusion of molecular tracers of size 40 kD and higher is reduced by 50% through its thickness [173]. It was shown that in the brain, the luminal surface of capillaries was 40% covered by glycocalyx, compared to 15% in the heart and 4% in the lungs; its thickness was also the highest in the brain (301 nm versus 136 nm in the heart, and 65 nm in the lungs) [174].

Many factors can lead to glycocalyx damage. For example, systemic inflammation in vivo leads to brain endothelial glycocalyx damage with decreased thickness and coverage, as shown in several studies following intravenous [175] or intraperitoneal [174] exposure to LPS in mice. Exposure to various factors such as LPS, TNF-α, or thrombin resulted in a significant reduction in glycocalyx thickness in human pulmonary microvessel endothelial cell lines under in vitro conditions [176], such that multiple circulating molecules during systemic infection or inflammation can lead to glycocalyx destruction [177]. 

GAGs are also a component of the glomerular basal membranes of the kidney (GBM). The GBM is an amorphous extracellular structure, between 300 and 350 nm thick, which was previously thought to be the main macromolecular filter, selective for size and charge. Normal GBM consists of laminin-521 (α5β2γ1), type IV collagen α3α4α5, nidogen, and heparan-sulphate proteoglycan (predominantly agrin). Podocytes are considered to be the main source of the GBM components laminin β2 and collagen network α3α4α5 [178]. Moreover, one of the key components of GBM is heparan sulphate. Due to its negative charge, heparan sulphate chains are highly hydrated and play a key role in the molecular unloading of GBM, although its direct role in filtration has been challenged by some researchers [179]. Damage and loss of any GBM components plays an important role in the pathogenesis of albuminuria, such as in DN [180]. Heparansulfate degradation induced by podocyte heparanase can occur in podocytopathies, leading to GBM damage [179]. As shown in our studies, acute single poisoning of rats with POX leads to increased urinary GAG excretion in rats, observed up to 12 weeks after poisoning. Apparently, even an episode of single OPs poisoning alters the levels of GAG synthesis and tissue localisation [60].

#### 2.4.2. Endothelial Glycosaminoglycans

Endothelial cells synthesise heparan sulphate, chondroitin, and dermatansulphate. In the glomerular basal membrane and mesangial matrix, GAGs have a faster turnover rate than in other tissues. The glycocalyx covering ECs is a matrix-like gel composed of proteoglycans, negatively charged and neutral glycosaminoglycans, glycoproteins, and plasma proteins [181]. The glycocalyx also contains hyaluronan, which is essential for maintaining a selective barrier for macromolecules. Disruption of the glycocalyx results in impaired GBM selectivity. Negatively charged components are abundantly present in this gel-like cell membrane, which is thought to constitute at least part of the charge barrier in the kidney and other organs. 

Endothelial loss of hyaluronan leads to impaired glomerular endothelial stabilisation. Hyaluronan of the glomerulus endothelium is a previously unrecognised key component of the extracellular matrix, which is essential for glomerular structure and function and is lost in DN. In 2019, a hyaluronan-specific probe was used to detect the loss of glomerulus endothelial hyaluronan in relation to the formation of lesions in the tissues of patients with DN. Loss of hyaluronan, which contains a specific binding site for angiopoietin and is a key regulator of endothelial resting and maintenance of EC barrier function, leads to impaired angiopoietin 1/Tie2 function [182].

Enzymes from the family of matrix metalloproteinases (MMPs) secreted by leukocytes or endothelial cells [183] can degrade endothelial glycocalyx components such as syndecan-4, as shown in human and mouse cell lines and rat mesenteric vessels [177,184,185]. MMPs such as MMP-2 and MMP-9 can actively participate in this process and are ultimately responsible for the development of albuminuria, AKI, and the activation of fibrosis. More details about the role of these matrix metalloproteinases in the pathogenesis of acute and chronic renal damage and glycocalyx destruction can be found in a recently published extensive review by Wozniak and co-authors [186]. 

It should be noted that there are rather few studies on the effect of OPs on pathophysiological processes involving MMPs. ROS have been shown to increase MMP-9, especially in rat brain tissue astrocytes [187]. Accordingly, as a result of increased oxidative stress, members of the MMP family also increase and alter extracellular matrix components such as laminin and fibronectin [188]. A 2020 study found that Notch and Notch2 activation via ADAM10 not only mediates proximal tubule development, but excessive activation of this pathway due to prenatal exposure to chlorpyrifos leads to the development of renal fibrosis in adult mice [189]. Analysis of human kidney biopsy specimens revealed a positive correlation between increased levels of ADAM10 and Notch2 and the development of fibrosis [186,190]. 

It is likely that there are also many factors that, when acting on endothelial cells, can provoke an increase in GAG synthesis. Increased GAG biosynthesis, according to some researchers, may enhance damage to the renal filtration barrier [181]. Proteinuria has previously been suggested to be caused by changes in the charge selectivity of GBM and/or the epithelial cell layer (podocytes). However, there is evidence that the endothelial lining of the luminal surface, consisting of proteoglycans with their associated glycosaminoglycan taps and glycoproteins, may contribute to permselectivity [181]. For example, increased GAG synthesis occurs when cells are stimulated by IL-1β. This response may be part of the cellular defence to maintain the permissive properties of the glomerular barrier, but there are differences between micro- and macrovascular endothelium as IL-1β stimulation of porcine aortic endothelial cells decreases GAG synthesis [181]. In a population of HUVEC, the effect of OPs such as tris (1, 3-dichloro-2-propyl) phosphate (TDCPP) was shown to reduce the expression of Nrf2, SOD1, and SOD2 in a dose-dependent manner [133]. Microscopically, a marker of increased GAG synthesis in ECs of glomerular capillaries, is the appearance of multiple vacuoles in the cell cytoplasm [191].

### 2.5. Immunological Mechanisms of OP Nephrotoxicity 

The immunomodulatory effect of OP metabolites is manifested in the disruption of cytokine secretion and the resulting altered immune cell response to antigenic stimulation [192]. OPs affect the immune response, including effects on antibody production; interleukin-2 production; T-cell proliferation; decreased CD5 cell count and increased CD26 cell count and autoantibodies; Th1/Th2 cytokine profiles; inhibition of natural killer (NK); lymphokine activated killer (LAK); and cytotoxic T lymphocyte (CTL) activity [193]. 

OPs inhibit the activity of NK cells, LAK cells, and CTL by at least the following three mechanisms: (1) OPs disrupt the granule exocytosis pathway of NK cells, LAK cells, and CTL by inhibiting granzyme activity and reducing intracellular levels of perforin, granzyme A, and granulisine, which is mediated by inducing NK cell degranulation and inhibiting the transcription of perforin, granzyme A, and granulisine mRNA; (2) OPs disrupt the FasLFas pathway of NK cells, LAK cells, and CTLs, which was investigated using perforin knockout mice, in which the NK cell granulosin exocytosis pathway is non-functional and only the FasLFas pathway remains functional; (3) OPs induce immune cell apoptosis [193]. 

Unfortunately, currently, there are no publications that consider the mechanisms of nephrotoxicity in acute OP poisoning from the perspective of immunotoxicity of these substances. Nevertheless, having information on changes in the system of cytokine regulation under the influence of influencing factors similar in terms of pathogenesis, one may notice some probable elements of this mechanism. In particular, it is known that such cytokines as transforming growth factor-β (TGF-β) is increased in DN and other kidney diseases [194]. Increased levels of TGF-β reduce the expression of megalin/cubilin and thus inhibit the internalisation of albumin [195]. This effect is dependent on the transcription factors Smad2 and Smad3. Exposure to OPs also stimulates the release of this cytokine. This has been shown in human THP-1 cell culture exposed to parathion, chlorpyrophos, and diazinon. The effect of OPs leads to the stimulation of increased cytokine release of TNFα, IL-1β, PDGF, and TGFβ mRNA [196]. Such facts bring our attention back to the understudied processes of reabsorption of OP-modified albumin in renal tubules mentioned in Section 2.1.2 above. 

A review was published recently on the immunotoxic properties of OPs in relation to the mechanisms of immune impairment under the influence of SARS-CoV-2 [197]. The authors postulate that, being lipophilic in nature, OPs crosses the cell membrane and passes through the phase I detoxification mechanism, generating excessive amounts of ROS. Excess ROS and subsequent oxidative stress is a major factor in immunotoxicity. ROS inhibits phosphatases, enhancing the production of pro-inflammatory cytokines. ROS also stabilises hypoxia-inducible factor 1-alpha (HIF-1α) by oxidising prolyl hydroxylase (PHD) to transcribe vascular endothelial growth factor (VEGF), which is involved in airway inflammation, airway hyper-responsiveness, and lymphocyte dysfunction. ROS induces lipid peroxidation, protein degradation, and DNA damage simultaneously. Protein degradation and DNA damage simultaneously trigger necroinflammation. OPs activate cytokine signalling suppressor-3 (SOCS3), disrupting the JAK/STAT-mediated antiviral immune response. Mitochondrial boundary disruption is facilitated by kinase 1 (ASK-1), which regulates apoptosis signalling and remains inactive as long as it is bound to thioredoxin. ROS oxidises thioredoxin, promoting apoptosome or inflammasome-mediated cell death [197]. A similar mechanism is likely to be involved in the most OP-vulnerable elements of the nephron, in particular the cells of the proximal tubules. Excess ROS and associated lipid peroxidation lead to disruption of intracellular protein synthesis, DNA damage, and initiation of apoptosis/necrosis of renal tubule epithelial cells. Cytokine dysregulation makes a certain contribution to these processes. OP poisoning has been shown to increase IL-2 production and T-cell proliferation; decrease circulating CD5 and increase CD26; and to stimulate autoantibody synthesis, alter Th1/Th2 cytokine profiles, and inhibit natural killer (NK), lymphokine activated killer (LAK), and cytotoxic T lymphocyte (CTL) activity [193]. Immune-mediated changes in renal cells are likely to depend on the specific OPs representative, its dose, and the duration of intoxication. However, it should be noted that the mechanisms of the consequences of acute OPs poisoning in the aspect of development of delayed nephrotoxicity phenomena caused by immunocompetent regulatory pathways are practically not studied.

### 2.6. Biochemical Mechanisms of OPs Nephrotoxicity

Since OPs possess different toxicokinetic characteristics, they have a different spectrum of molecular and cellular targets, and among the factors on which their selective action depends, the main place belongs to their metabolic transformations in the organism [23,198,199]. Nevertheless, there are general trends of changes in the complex biochemical processes. Along with their prooxidative effect, OPs cause changes in protein metabolism via phosphorylation and oxidation of proteins and disrupt not only cholinesterases but also several other enzymes such as protein kinases, ATP-ase, trypsin, succinate dehydrogenase, phospholipase C, and others [8,198]. 

On the basis of polytropic action of OPs, causing disorders of several enzyme systems, the activity of aminotransferases, gamma-glutamyltransferase, and paraoxonase-1 were proposed as additional biochemical markers of OP intoxication [200,201], which were shown to be the biochemical markers of intoxication during the first weeks after acute poisoning by highly toxic OPs. 

In rats, after malathion poisoning, there is an increase in functional renal tests, serum creatinine, urea, and uric acid [202,203]. An increase in creatinine and urea concentrations in plasma or serum of rats after OPs poisoning at different doses has been observed for many members of this group of chemicals, at different times of observation and with different routes of administration. In particular, this has been described for methyl parathion at a dose of 0.56 mg/kg (1/25 LD50) per 3 days p/o for 8 weeks [40]; soman at a single dose of 0.67 LD50 [204]; malathion at a dose of 75 mg/kg daily p/o for 7 days [205] and at a dose of 100 mg/kg daily p/o for 30 days [35]; diazinon once intraperitoneally at a dose of 100 mg/kg [33]; chlorpyrophos at a dose of 10 mg/kg combined with carbendazim at 50 mg/kg p/o for 7 days [206]; and some others. Thus, it can be said that potential nephrotoxicity has been established for many OPs if increased creatinine and urea concentrations are taken as evidence of impaired renal function.

In our studies after acute poisoning of rats with POX, a decrease in endogenous creatinine clearance (ECC) within 24 h was found, irrespective of the method of carboxylesterase inhibition. At 3 and 7 days after poisoning, ECC in poisoned rats did not differ from those of intact controls [61]. This may indicate a decrease in GFR in the early period after poisoning, with subsequent recovery of filtration levels. The decrease in effective RBF and GFR in rats after administration of POX at doses causing systemic toxicity was demonstrated in 1970 [207]. The decrease in ESS after OP poisoning has also been registered in humans, e.g., in the case of diazinon [208] and obidoxime [209].

It should be noted that OP poisoning also causes metabolic disorders, leading to hyperglycaemia. Hyperglycaemia and glycosuria were observed in the assessment of the glycaemic status of patients with OPs poisoning, and 57% of cases of hyperglycaemia were observed in the severe form of poisoning. With increasing severity of poisoning, plasma glucose and MDA levels increase, and serum cholinesterase levels decrease [210]. Hyperglycaemia leads to increased non-enzymatic glycation by binding glucose or by-products to amino groups of proteins and forming complex compounds, glycation end products (AGEs), which alter the structure and function of proteins. Glycated proteins activate specific receptors for glycation-expanded end products (RAGE) and cause intracellular oxidative stress [129]. Thus, hyperglycaemia is one of the factors of oxidative stress mechanism in OP poisoning [134]. In our studies in rats, 24 h after acute poisoning with POX, glucosuria was observed without an increase in blood glucose level, indicating its renal origin [61]. It should be noted that glycosuria without associated hyperglycaemia after OP poisoning has also been described in humans [211,212]. The aetiology of such glycosuria is not well understood, but renal tubular damage is thought to be a factor in pathogenesis [211,213]. 

## 3. Histological and Ultrastructural Changes in the Kidneys

Each representative of OPs has a unique toxicity profile, and therefore nephrotoxicity and histopathological changes in the kidneys may differ when poisoned by different OPs. In the last few years, isolated reports of observed changes in renal morphology when exposed to OPs, such as malathion, fentione, diisopropylfluorophosphate (DFP), and paraoxon, continue to be published [32,40,53,60,61]. The pathomorphophysiological changes in renal tissue in OP-treated rats include shrinkage of glomeruli; dilation of the urinary space; infiltration of interstitial tissue with inflammatory cells; pycnotic nuclei; vacuolisation of renal tubule cytoplasm; deposition of hyaline material in the lumen of some tubules; dilation and occlusion of blood vessels; degeneration of renal tubule epithelium; deposition of epithelial casts in some tubule lumen; and tearing of Bowman capsule [61,203]. 

Recently, it was pointed out by some authors that the mechanisms of pathogenesis of renal tissue after exposure to OPs are not fully established, and there are many speculative assumptions in the limited literature [214]. Some of the proposed mechanisms are related to pseudocholinesterase levels in the distal diffuse renal tubule and oxidative stress due to high intratubular OPs concentration, rhabdomyolysis, and hypovolemia due to dehydration [215,216]. Renal circulation and electrolyte excretion are also thought to be partly controlled by a cholinergic mechanism, so OP poisoning impairs renal function [27]. 

Despite limited knowledge of the pathophysiology, the histopathological damage caused by OPs is better understood. In nephrons of rats poisoned by impurities and by-products of malathion, researchers have revealed swelling, deformation, and distension of glomeruli, as well as narrowing of the initial part of the proximal tubule [217]. In chlorpyrifos poisoning in rat kidneys, there was a reduction, degeneration, and hypercellularity of glomeruli, enlargement of tubules, hypertrophy of epithelium and degeneration of renal tubules, deposition of eosin-positive substances in glomeruli and renal tubules, and infiltration by leukocytes. However, the intensity of these changes varied according to the dose and duration of exposure [218]. After daily poisoning of rats with ekalux insecticide at doses of 1/20, 1/30, and 1/40 of 19.95 mg/kg (LD50) for 10 days, cellular debris occlusion of the capillary lumen was observed in the kidneys. The basal membrane is irregularly wrinkled and branched and contains multiple large electron-dense conglomerates [39]. An acute poisoning with POX in a dose of LD84 leads to an increase in GBM thickness in 7 days, although accumulation of electron-dense deposits in it is not observed [61].

After acute OP exposure, the epithelium of the convoluted tubules is initially damaged [32,38,39]. Damage to tubules after OP intoxication is the main cause of renal dysfunction [32,41]. In a morphological study, intoxication with a single sublethal dose of POX revealed disorganisation and moderately pronounced apoptosis of tubule epithelial cells with lumen distension. The authors attributed the findings to the direct effect of OPs on cells provoking oxidative stress leading to damage to renal tubules and renal parenchyma, dehydration due to hypovolemia, and development of acute renal damage [32]. 

## 4. Conclusions

The molecular signalling pathways leading to cell survival or death after exposure to OPs are poorly understood. Researchers agree that OPs can activate signalling pathways for oxidative stress through the production of ROS, leading to further tissue damage through the induction of lipid peroxidation, protein oxidation, and cell apoptosis [133]. The role of oxidative stress in the pathophysiology of poisoning for various OPs is difficult to generalise. Mechanisms other than lipid peroxidation, including ATP depletion, DNA damage, protein oxidation, and increased intracellular calcium due to impaired cytoplasmic membrane permeability, are also involved in the process of cell death [126]. The presence of multiple apoptosis and mitochondrial destruction in cells constituting the filtration barrier and renal parenchyma has been noted by all authors investigating the morphological manifestations of the effects of OPs poisoning in renal tissues [32,38,39]. 

In conclusion, molecular mechanisms of OP nephrotoxicity discussed in this review can serve as a basis for theoretical assumptions and speculation. Unfortunately, there is very little information on the pathogenesis of delayed effects of acute OP poisoning such as renal damage. The vast majority of published works refer to the observation and description of effects in subchronic and chronic poisoning with these compounds. We hope that our attempt to take a view on the molecular mechanisms of OP nephrotoxicity will prove useful in planning new research in this area.

## Figures and Tables

**Figure 1 ijms-23-08855-f001:**
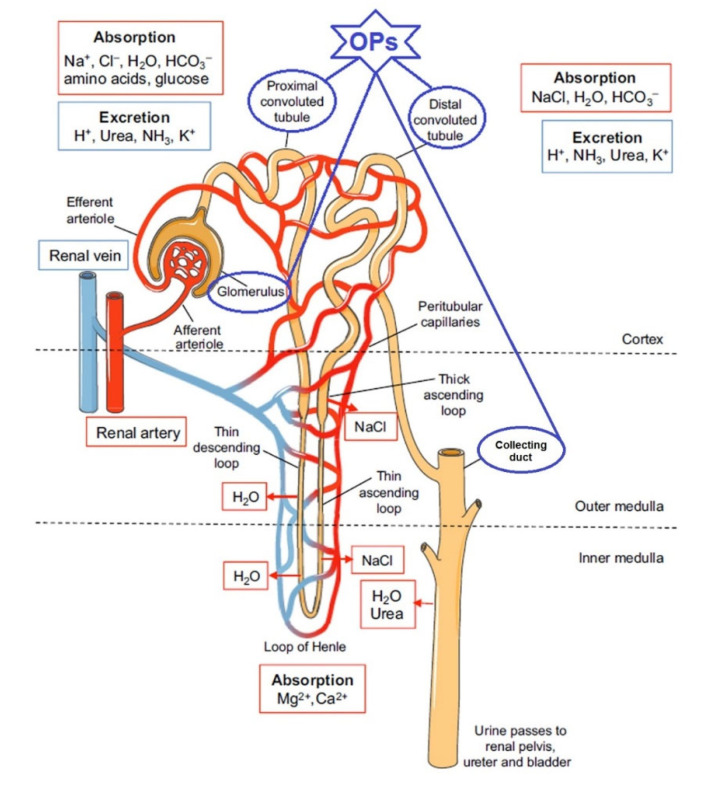
Schematic of a nephron. This schematic shows a nephron, the functional unit of the kidney. The most vulnerable elements of the nephron for OPs are shown in oval frames. Blood is delivered to the glomerulus, where plasma is filtered into the lumen of the tubule. Various ions are excreted and absorbed, and water is retrieved, as plasma passes through the different segments of the tubule, which are intimately linked to peritubular capillaries. Concentrated urine is formed by this filtration process, which then passes through the collecting duct to the renal pelvis. The different components of a nephron occupy distinct regions of the kidney: the cortex and outer and inner medulla, as shown (after [44] with additions). Creative Commons Attribution License.

**Figure 2 ijms-23-08855-f002:**
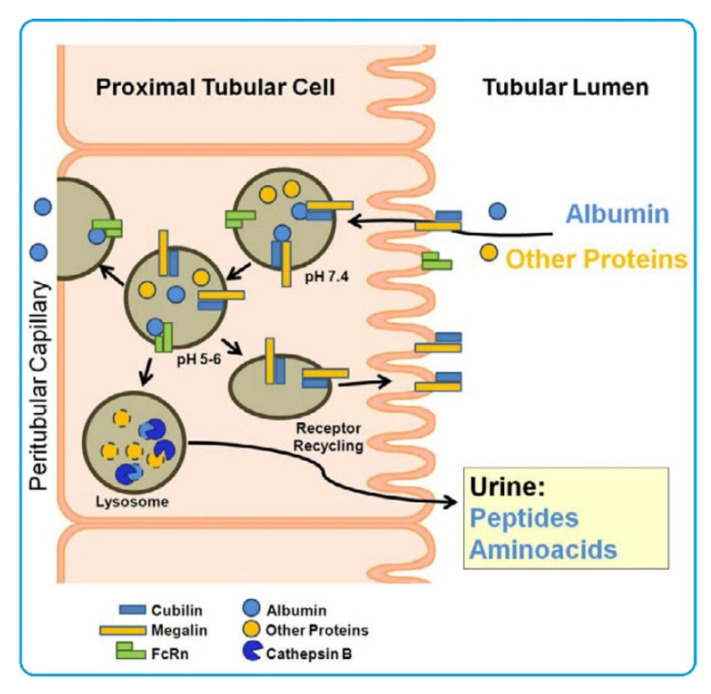
Albumin (blue) and other proteins (yellow), after binding to megalin/cubilin complex on the brush border of proximal tubular cells, are internalised, via clathrin-coated vesicles, that fuse to early endosomes. Acidification of these endosomes causes dissociation of albumin from the megalin/cubilin complex, but, at pH 5–6, albumin binds to FcRn and undergoes transcytosis. The megalin/cubilin complex is recycled to the apical brushborder membrane, while unbound albumin and other proteins are transferred to lysosomes for degradation by cathepsin B and other lysosomal proteases. Degradation products are exocytosed into the tubular lumen and excreted in the urine. (after [85]). Creative Commons Attribution License.

**Figure 3 ijms-23-08855-f003:**
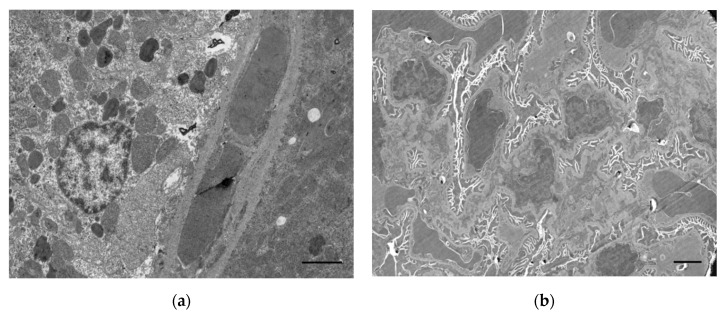
Ultrastructural changes in the kidney of a rat that died due to acute paraoxon poisoning at a dose of LD84. (**a**) Stasis in the blood capillaries of the interstitium and preservation of the mitochondria of the tubular epithelial cells; (**b**) spastic contraction and stasis in the glomerular capillaries of the renal cortical layer. Scale bar = 2 μm.

**Figure 4 ijms-23-08855-f004:**
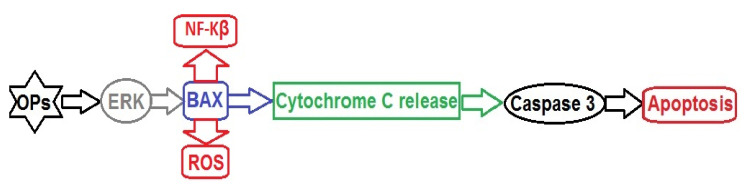
OP-induced nephrotoxicity pathway mediated by MAPK signalling. Abbreviations: OPs: organophosphates, ERK: extracellular-signal-regulated protein kinase, NF-Kβ: nuclear transcription factor kappa-β, ROS: reactive oxygen species.

**Figure 5 ijms-23-08855-f005:**
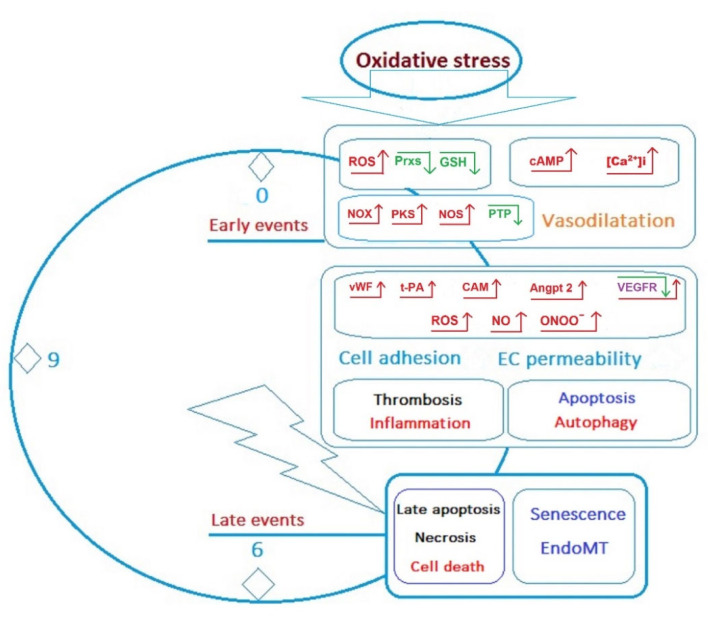
Early and late events developed in ECs upon the influence of oxidative stress. Early changes in EC under the influence of oxidative stress and increased ROS involve a cascade of biochemical reactions leading to the development of vasodilation. In the longer term, especially under conditions of chronic oxidative stress, there is an increase in cell adhesion and endothelial permeability disorders, eventually leading to induction of cell senescence and/or endothelial to mesenchymal transition (EndoMT), apoptosis, development of inflammatory/necrotic processes, and cell death. ↑—increase; ↓—decrease.

**Table 1 ijms-23-08855-t001:** Endothelial cell functions.

Physiological Functions	Regulatory Factors
Leukocyte trafficking	ICAM1, VCAM, E-selectin
Inflammation	MCP1, complement activation, oxidative stress
Metabolism	Glycolysis, glutamine and asparagine metabolism, FAO, mTOR, HIF
Vascular permeability and glomerular filtration	Endothelial fenestrations, glycocalyx, VEGF, CD146
Vascular tone	Vasodilators: NO, H_2_S, PGI_2_ Vasoconstrictors: endothelin, TXA_2_, PGH_2_
Haemostasis and coagulation	Anticoagulant factors: TM, glycocalyx, tPA, TFPI, PGI_2_ Procoagulant factors: vWF, TF, TXA_2_, PAI1
Control of VSMC proliferation	Microparticles, Jagged 1–NOTCH-1, miR-126, NO
Angiogenesis	VEGF/VEGFR, angiopoietin 2/TIE2, HIF activation, CD146, glycolysis, glutamine and asparagine metabolism, FAO, vWF, Jagged 1/NOTCH-1

## Data Availability

The data presented in this study are available on request from the corresponding author.

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
