# Peer review of "Molecular Mechanisms of Acute Organophosphate Nephrotoxicity"

_ijms, 2022, doi:10.3390/ijms23168855_

Round 1
Reviewer 1 Report
The authors summarize current knowledge of organophosphates (OP) associated with nephrotoxicity and potential molecular mechanisms underlying the toxicities. The authors briefly describe factors that could affect the outcomes of OP exposures such as routes of OP exposures, their kinetics, duration of exposures, etc. The authors also provide rationale for writing this review article.
Comments/suggestions:
- Adding structure of albumin and how it interacts with OP would be helpful.
- Please include a brief description for Figure 5.
- Page 16 third paragraph, “A similar mechanism is likely to be involved in the most OPs-vulnerable elements of the nephron, in particular the cells of the proximal tubules.” Need to elaborate more and how the changes would cause damage to the kidneys.
Minor issues:
- Delete “20” from page 6, second paragraph from the bottom, “…Megalin and cubulin are among the 20 most…”
- Page 10, third paragraph from the bottom, malone dialdehyde (MDA) is misspelling and it has already spelled out in page 8 (last sentence).
- Page 13, last paragraph, advance glycation end products (AGE) has already spelled out on page 9.
Author Response
We are extremely grateful to the reviewers, their high appreciation of our work, and comments, thanks to which we were able to improve the quality of the review. Below our responses to the notes of the 1st reviewer.
Reviewer 1
- Adding structure of albumin and how it interacts with OP would be helpful.
1) The structure of albumin and the features of its interaction with OPs according to molecular modeling data on the example of paraoxon were repeatedly presented by us in earlier works, some of which are cited under Nos. 63 and 78 in the bibliography:
- Goncharov, N.V., Belinskaia D.A., Shmurak V.I., Terpilowski M.A., Jenkins R.O., Avdonin P.V.Serum Albumin Binding and Esterase Activity: Mechanistic Interactions with Organophosphates. Molecules. 2017, 22 (7): 1201. DOI: 10.3390/molecules22071201
- Belinskaia, D. A., Terpilovskii, M. A., Batalova, A. A., & Goncharov, N. V. (2019). Effect of Cys34 Oxidation State of Albumin on Its Interaction with Paraoxon according to Molecular Modeling Data. Russian Journal of Bioorganic Chemistry, 45(6), 535–544. doi: 10.1134/s1068162019060086
We consider it incorrect to represent graphically a generalized scheme of the interaction of OPs with albumin due to the peculiarities of these interactions for different OPs.
- Please include a brief description for Figure 5.
Inserted
- Page 16 third paragraph, “A similar mechanism is likely to be involved in the most OPs-vulnerable elements of the nephron, in particular the cells of the proximal tubules.” Need to elaborate more and how the changes would cause damage to the kidneys.
Inserted.
Minor issues:
- Delete “20” from page 6, second paragraph from the bottom, “…Megalin and cubulin are among the 20 most…”
Corrected.
- Page 10, third paragraph from the bottom, malone dialdehyde (MDA) is misspelling and it has already spelled out in page 8 (last sentence).
Corrected.
- Page 13, last paragraph, advance glycation end products (AGE) has already spelled out on page 9.
Corrected.
Reviewer 2 Report
I appreciate the opportunity to review the manuscript:
“Molecular mechanisms of acute organophosphate nephrotoxicity” Vladislav E. Sobolev, Margarita O. Sokolova, Richard O. Jenkins and Nikolay V. Goncharov
The manuscript presented for evaluation is very interesting and relates to a clinically useful topic. Organophosphates (OPs) are toxic chemicals produced by an esterification process and some other routes. They are the main components of herbicides, pesticides and insecticides and are also widely used in the production of plastics and solvents. Acute or chronic exposure to OPs can manifest in various levels of toxicity to humans, animals, plants and insects. OPs containing insecticides were widely used in many countries during the 20th century and some of them continue to be used today. The manuscript has a typical layout and is correct for review papers. The manuscript is written in correct English, the summary properly presents the most important issues raised in the manuscript, the figures have been prepared correctly, the literature is presented correctly. Manuscript's editorial level is formally and stylistically correct.
Recommendation: Manuscript may be published in its current form.

Author Response
We are extremely grateful to the reviewers, their high appreciation of our work, and comments, thanks to which we were able to improve the quality of the review.